# Factors Influencing Battery Electric Vehicle Adoption in Thailand—Expanding the Unified Theory of Acceptance and Use of Technology's Variables

**Phasiri Manutworakit [1] and Kasem Choocharukul [2],***

[1] Interdisciplinary Program Environment, Development and Sustainability Program, Graduate School, Chulalongkorn University, Phyathai Road, Wangmai, Pathumwan, Bangkok 10330, Thailand; m.phasiri@gmail.com

[2] Department of Civil Engineering, Faculty of Engineering, Chulalongkorn University, Phyathai Road, Wangmai, Pathumwan, Bangkok 10330, Thailand

* Correspondence: kasem.choo@chula.ac.th

**Abstract:** The replacement of combustion cars with battery electric cars can help to support sustainable transport and renewable energy in green transportation in Thailand; however, the diffusion rates of electric cars are still low. The purpose of this study is to examine factors that influence Thai car owners' adoption of battery electric vehicles. The proposed model expands the Unified Theory of Acceptance and Use of Technology (UTAUT) as the theoretical framework. Data were collected through an online questionnaire survey completed by 403 participants in Bangkok and the vicinity and analyzed using partial least squares structural equation modeling (PLS-SEM). The result showed that purchase intention is significantly and positively influenced by performance expectancy, effort expectancy, social influence, hedonic motivation, and environmental concern. In contrast, purchase intention is not significantly influenced by price value and policy measures. Use behavior is positively influenced by purchase intention. Facilitating conditions do not significantly influence purchase intention and use behavior. Moreover, only the age variable was found to have significant effects on purchase behavior. This study suggests that governments with incentive policies and electric car manufacturers should focus on improving cars to increase battery electric car adoption.

**Keywords:** battery electric vehicle; sustainable transport; green transportation; the unified theory of acceptance and use of technology; purchase intention and behavior; Thailand

## 1. Introduction

The transport sector causes energy usage and air pollution. Many cities face energy consumption problems and air pollution due to the increasing number of cars. This is a common challenge for cities all over the world, particularly in developing countries such as Thailand. The use of automotive technology, powered by electric vehicles, is a useful sustainable transport option for energy saving and reducing pollution [1–3]. People should consider the electric vehicle because it releases less greenhouse gas emissions than the combustion engine vehicle.

China is the largest market of electric cars in the world, with electric cars constituting nearly half of all existing electric cars. The first market share of electric cars in the world was in Norway, and many Norwegians buy electric cars instead of combustion cars [4–6]. The electric vehicles in the current automotive market can be divided into four main types, namely, the Hybrid Electric Vehicle (HEV), Plug-in Hybrid Electric Vehicle (PHEV), Battery Electric Vehicle (BEV) and Fuel Cell Electric Vehicle (FCEV) or combustion vehicle. In comparison, the HEV still relies on fossil fuels but is more efficient than normal engines. The PHEV can be charged from the vehicle itself running over a distance but within a limited weight. It has both a combination of the engine and electrical system. While the

BEV uses only electrical energy, meaning it emits zero emissions, the energy efficiency of the BEV is higher. More carbon monoxide, nitrogen oxides, hydrocarbons and particular matter are released when fuel is burnt in the FCEV than in other vehicles. The FCEV takes a long time to refuel and is a new technology with a high cost. By studying the overall energy consumption, the BEV is the most efficient energy usage, and it is probably the best choice for sustainable transport [7].

The electric vehicles are a trend around the world at present. Car markets such as China, the USA, the United Kingdom, Norway, Germany, the Netherlands, Sweden, and France [8], are rapidly growing. On the other hand, the market of electric vehicles in Thailand was initially introduced to consumers in the personal car market. An increasing number of electric vehicles in many countries can reduce environmental problems and energy consumption because the pollutants of electric vehicles are lower than those of combustion vehicles [7]. Hence, the research papers in this area are essential to explore the factors and issues that stimulate electric vehicle adoption [9–11].

Hence, many countries have to plan and create policies to support battery electric vehicle adoption in order to achieve the Sustainable Development Goals: improving energy efficiency (Goal 7); the need to make cities inclusive, safe, resilient and sustainable (Goal 11); and climate action (Goal 13). In the case of Thailand, the promotion of electric vehicles actively begun in early 2015 by the Reform National Council and government at the time of General Prayuth Chan-o-cha, which has been a major administration in driving electric vehicles in Thailand, among others, including the Ministry of Energy, the Ministry of Industry, the Ministry of Higher Education, Science, Research and Innovation and Ministry of Transport. There is a target to reduce the greenhouse gas emissions in the road sector by 25% in all regions by 2030. Electric vehicles appear in many levels of development plans, including the 12th National Economic and Social Development Plan (2017–2021), the Energy Efficiency Plan (2015) and the 20 years strategic plan for sustainable transport development. Based on the Department of Land Transport (DLT) database from 2015 to 2019, the number of electric cars in Thailand steadily increased, but diffusion rates of electric cars remained low [12]. The annual target 2035 for electric cars and pickup trucks is 1.154 million units. The accumulative target 2035 for electric cars and pickup trucks is 8.75 million units. Electric car usage and emission targets are currently far out of reach. Obviously, the adoption of electric cars would benefit not only the user, but also the environment.

In prior studies related to the adoption intention and behavior related to electric vehicles, researchers have investigated factors using different methods [13–17]. The theory of planned behavior (TPB) is the most widely used. They found that subjective norm, attitude, and perceived behavioral control have different relationships with intention and behavior [13,17]. In addition, environmental awareness and incentive policies will have direct effects on the electric vehicle purchase intention [14,15]. Some research has improved the consumption value theory and perceptions of electric vehicle adoption, which are divided into functional values and non-functional values [16]. The functional values of electric cars such as convenience, performance, and low cost, have direct and indirect effects on the adoption of electric vehicles. Non-functional values such as social responsibility, and emotional wellbeing, have only indirect effects on the adoption of electric vehicles. They are considered factors that impact the purchase intention and behavior of users, but which are insufficient to explain individual purchase intention and behavior. Venkatesh et al. developed the unified theory of acceptance and use of technology (UTAUT) from existing theories of behavior, which is made up of key constructs such as performance expectancy, effort expectancy, social influence, and facilitating conditions, and it applies gender, age, experience, and voluntariness of use which are moderated key constructs [18]. New technology studies have been conducted using the UTAUT model. Researchers have found new constructs, such as hedonic motivation, price value, and environmental concern, to study purchase intention and use behavior. Moreover, policy measures encourage people to adopt electric vehicles, especially in developed countries. Pricing incentives and increasing charging stations will be the best policies to adopt electric cars [14]. However,

the UTAUT model has rarely seen the adoption of battery electric vehicles. In the case of Thailand, there is a scarcity of research on the battery electric vehicle adoption of Thai respondents. This paper focused on Thailand's private and public sectors in order to promote the further development of electric cars in Thailand. Therefore, this study seeks to empirically investigate factors related to purchase behavior in the adoption of battery electric vehicle by using the extended UTAUT model. The findings of this paper could provide guidance for the government and suppliers to take measures for Thailand.

The organization of this paper is as follows. Section 2 provides the background of the UTAUT model and an overview of the factors influencing battery electric vehicle adoption worldwide. Section 3 presents the research hypotheses, materials, and methods used in the research. Section 4 explains the results from the data collected, verifies data validity and reliability, tests the hypothesized model, and analyses policy measures. Section 5 elaborates on the main discussions from this study's findings and policy implications. Finally, Section 6 provides a conclusion, limitations, and future research directions.

## 2. Literature Review

In this study, the unified theory of acceptance and use of technology (UTAUT) was applied to examine the effect of purchase intention on BEVs. First, the background of the UTAUT model is described, followed by its related constructs. Second, an overview of factors influencing battery electric vehicle adoption in various countries around the world is illustrated.

### 2.1. Background of UTAUT Model

The unified theory of acceptance and use of technology (UTAUT) is the one of the models that was developed from eight existing theories of behavior, namely, the Theory of Reasoned Action (TRA), the Technology Acceptance Model (TAM), the Theory of Planned Behavior (TPB), the Motivational Model (MM), the Combined TAM-TPB Model, the Model of Personal Computer Utilization (MPCU), the Diffusion of Innovations (DOI) and Social Cognitive Theory (SCT) [19–21]. This model has four main factors of behavioral intention (BI), including performance expectancy (PE), effort expectancy (EE) and social influence (SI). Facilitating conditions (FC) became the factor that related to usage behavior. Moreover, gender, age, experience, and voluntariness of use constituted optional variables.

The model can be applied to the public acceptance of new technology transport modes such as electric bicycles, automated road transport systems and electric carsharing acceptance [22–24]. This study examined the impact of hedonic motivation, which has a positive impact on behavioral intention. Price value incorporated one of the other variables that was developed for individual acceptance and use setting [25]. Eco-friendly declaration for EVs and environmental action proposals can reinforce each other to help increase intention to purchase EVs [26,27]. Moreover, electric vehicle adoption in the world is highly dependent on strong electric vehicle policies, such as those in California, China, Germany, and Norway [28,29]. In regard to policies, prior research tends to examine incomprehensive policy instruments in promoting EV adoption, but policies are integrated both for financial and non-financial instruments [30–32]. In Korea, researchers found that environmental concerns and financial incentives impact adoption electric vehicle intention [33]. A German survey found that external policies, infrastructure, incentives, and communication were related to electric vehicle intention [34]. Their results showed that the main policies, including purchase subsidies, parking fee reductions, and driving privileges have a positive impact on consumer purchase intention [35–38]. Accordingly, hedonic motivation, price value, environmental concern, and policy measures (PM) take more into consideration than the factors from the original UTAUT.

### 2.2. Overview of Factors Influencing Battery Electric Vehicle Adoption

Many researchers have studied various factors influencing battery electric vehicle purchase intention, including demographic characteristics, car performance, psychological and social status needs, government policies, cost, environmental concerns, and facilities [9–11]. The key factors influencing battery electric vehicle adoption worldwide are the price, performance, usage cost, and time cost. In the United Kingdom, the performance considerations such as noise, comfort, ease of driving, and driving long distances, that showed good performance were the most critical factors affecting the adoption of battery electric vehicles. One factor that affects popularity is the battery life of electric vehicles; therefore, this problem must be resolved. This becomes a stronger purchase intention for battery electric vehicles.

## 3. Materials and Methods

To test the hypotheses, the methodology in this research was a quantitative survey of Thai car owners who were interested in BEVs. The data were analyzed using partial least squares structural equation modeling (PLS-SEM) through a program called Warp PLS 7.0 developed by Kock [39]. Warp PLS is a leading PLS-SEM software program with various features that are not found in other programs. It has the ability to unambiguously identify non-linear functions linking latent variables in SEM into pairs and to calculate multivariate coefficients of association.

### 3.1. Research Hypotheses

Combined with the previous section and the original UTAUT, a detailed description of our proposed UTAUT research model is provided in the following.

#### 3.1.1. Performance Expectancy (PE)

Venkatesh et al. [18,25] defined performance expectancy as the level of personal belief that using collaboration technology will improve work efficiency and lead to operational success. Electric cars are associated with many benefits, such as reduced energy use and air pollution.

**Hypothesis 1 (H1).** *Performance expectancy has a significant positive effect on BEV purchase intention.*

#### 3.1.2. Effort Expectancy (EE)

Effort expectancy refers to the level of awareness of the ease of using technology, or that it can be easily learned and used, is convenient, and is not complicated. The perception of simplicity allows users to anticipate technology performance and ultimately intend to demonstrate technology behavior. EE is applied to perceive ease of use in TAM.

**Hypothesis 2 (H2).** *Effort expectancy has a significant positive effect on BEV purchase intention.*

#### 3.1.3. Social Influence (SI)

The role of social influence arises from the individuals who influence the decisions of users such as their family and friends. Social influence is also defined as the power of a co-worker or supervisor to influence how technology users express themselves.

**Hypothesis 3 (H3).** *Social influence has significant positive effects on BEV purchase intention.*

#### 3.1.4. Facilitating Conditions (FC)

Facilitating conditions are defined as the availability of technology, organizational systems, and resources in terms of infrastructure, software system, and experts that the organization has prepared to support the use of technology. Moreover, facilitating conditions became the factor related to usage behavior.

**Hypothesis 4a (H4a).** *Facilitating conditions significantly positively affect BEV purchase intention.*

**Hypothesis 4b (H4b).** *Facilitating conditions have a significant positive effect on use behavior.*

3.1.5. Hedonic Motivation (HM)

Hedonic motivation is the fun or enjoyment derived from using technology. Perceived enjoyment impacts consumer acceptance and use of a new technology.

**Hypothesis 5 (H5).** *Hedonic motivation have a significant positive effect on BEV purchase intention.*

3.1.6. Price Value (PV)

Many research and social roles mentioned that price influences purchase intention [25]. Similarly, business owners operate at a lower cost and generate more profits or customers decide to buy cheap and good-quality products.

**Hypothesis 6 (H6).** *Price value has a significant positive effect on BEV purchase intention.*

3.1.7. Environmental Concern (EC)

With increasing global issues, environmental concerns have become more significant in purchasing decisions. Global warming from $CO_2$ emissions produced by cars is impacting the purchasing decisions of car consumers [40].

**Hypothesis 7 (H7).** *Environmental concern has a significant positive effect on BEV purchase intention.*

3.1.8. Policy Measures (PM)

Incentive policy measures are essential factors that influence purchase intention. If governments do not support EVs, consumers may have low intentions of purchasing EVs [41,42]. Incentive policy measures are divided into two categories: monetary and non-monetary incentive policy measures. Tax credit, subsidies, discount car price, exemption of new registered car tax, reduced parking fee, and free charging fees are examples of monetary incentive policy measures in developed countries. Non-monetary incentive policy measures aim to provide convenience to consumers when they buy and use BEV, such as a fast lane for EVs.

**Hypothesis 8 (H8).** *Policy measures have a significant positive effect on BEV purchase intention.*

3.1.9. Purchase Intention (PI) and Use Behavior (UB)

Perceived attitudes and use behavior determine actual actions [43]. Purchase intention is a major determinant of use behavior. The following hypothesis was proposed:

**Hypothesis 9 (H9).** *BEV purchase intention has a significant positive effect on user behavior.*

The conceptual model consists of 9 hypotheses and is shown in Figure 1.

*3.2. Sample and Data Collection*

The sample for this research included car owners living in Bangkok and the vicinity of Thailand who were over 18 years old and were interested in BEVs. The vicinity of Thailand encompasses the five adjacent provinces of Nakhon Pathom, Pathum Thani, Nonthaburi, Samut Prakan, and Samut Sakorn. Based on the Department of Land Transport (DLT) database, in 2020, the number of private cars (not more than 7 passenger seats) in Thailand was 559,553 cars [44]. The number of required samples is 400 respondents in accordance with the formula of Yamane, with a confidence interval of 95% and an allowable error of $\pm 5\%$ [45]. Additionally, an appropriate pilot study sample size is 10% of the sample size anticipated for the parent study [46]. A sample size for the pilot study of 40 respondents was adopted accordingly.

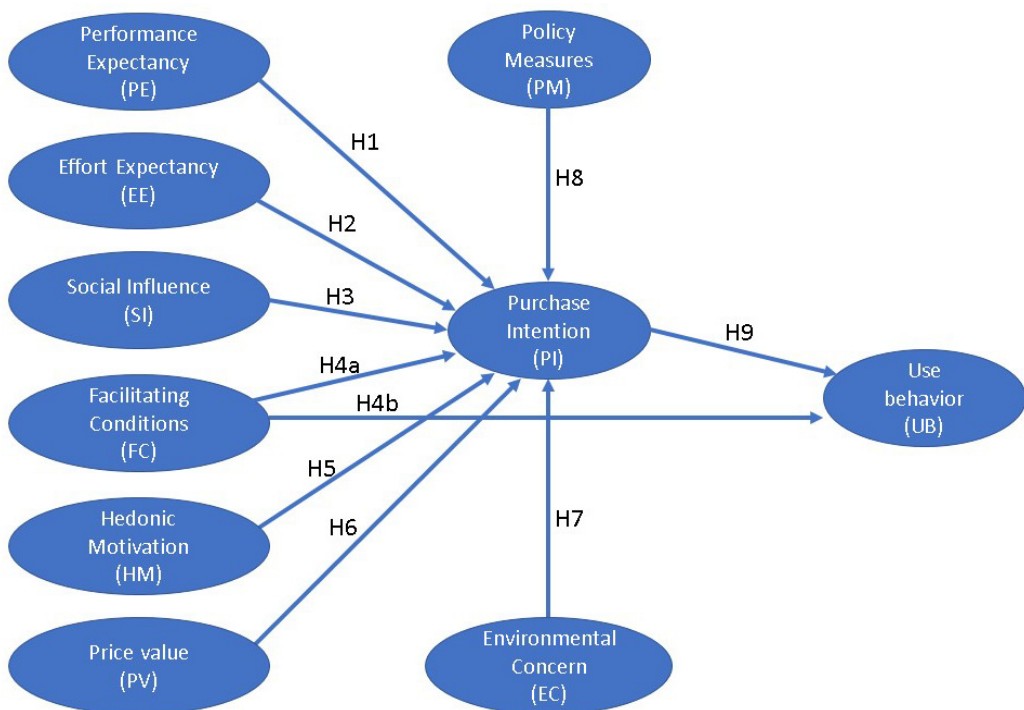

**Figure 1.** Conceptual model.

The survey questionnaire was first trialed with 40 car owners in Bangkok and the vicinity to gain feedback on the survey content. After that, the questionnaire was revised for its clarity and suitability. Due to the COVID-19 situation, the main study was conducted through online survey channels from October 2021 to December 2021, with a total of 412 responses received. The data were screened for missing data and exclusion criteria (do not own a car), leaving a final data set of n = 403 for analysis. Moreover, all subjects gave their informed consent for inclusion before they participated in the study. The study was conducted in accordance with the Declaration of Helsinki, and the protocol was approved by the Ethics Committee of Research Involving Human Subjects, and the second Allied Academic Group in Social Sciences, Humanities and Fine and Applied Arts at Chulalongkorn University.

*3.3. Measures of Constructs*

In this study, the statements or constructs in the survey questionnaire were adapted from various studies in the literature that applied the UTAUT as the main instrument, in which a pre-test is required to ensure reliability and validity. The questionnaire was divided into three sections: a demographic description of respondents, and the UTAUT containing 55 questions and consisting of ten constructs (performance expectancy (PE), effort expectancy (EE), social influence(SI), facilitating conditions (FC), hedonic motivation (HM), price value (PV), environmental concern (EC), policy measures (PM), purchase intention (PI), and use behavior (UB)). The 55 items were measured using a five-point Likert scale ranging from "strongly disagree = 1" to "strongly agree = 5". These constructs were modified to suit this study, and a summary of each variable is shown in Table 1. The final part, regarding policy measures, suggested that respondents reflected that Thailand should support more people who use BEVs.

**Table 1.** Example statements for ten constructs of the model.

| Constructs | Source (S) |
|---|---|
| Performance Expectancy (PE)<br>PE1 (I would find a BEV useful for my travel.)<br>PE2 (I think using a BEV would help make my travel more convenient.)<br>PE3 (I think using a BEV would reduce my energy cost per month.) | [18,47] |
| Effort Expectancy (EE)<br>EE1 (I would find a BEV easy to use.)<br>EE2 (I can learn to use it easily and quickly.) | [18,47] |
| Social Influence (SI)<br>SI1 (Social trends influence the decision to buy a BEV)<br>SI2 (I often explore what products others buy or use.) | [18,47] |
| Facilitating Conditions (FC)<br>FC1 (Example of the resources necessary to use a BEV are charging stations and service centers.)<br>FC2 (I have the knowledge necessary to use a BEV) | [25] |
| Hedonic Motivation (HM)<br>HM1 (Driving a BEV is fun and enjoyable.)<br>HM2 (Due to its smoothness and high acceleration, driving a BEV is very entertaining.) | [25] |
| Price Value (PV)<br>PV1 (The price of a BEV is an important factor for buying.)<br>PV2 (BEVs are reasonably priced.) | [25] |
| Environmental Concern (EC)<br>EC1 (I want to buy a BEV due to the air pollution crisis.)<br>EC2 (BEVs contribute to saving the environment for the next generation.) | [40] |
| Policy Measures (PM)<br>PM1 (Satisfaction with monetary incentive policy measures such as tax exemption, purchase subsidy, parking fee reduction and free charging fee.)<br>PM2 (Satisfaction with non-monetary incentive policy measures such as the right to use bus lanes and separate allocations of EV license plates.) | [48] |
| Purchase Intention (PI)<br>PI1 (If I had a BEV available, I would prefer to drive it rather than a traditional car.)<br>PI2 (If I had the chance, I would buy a BEV.) | [29,46] |
| Use Behavior (UB)<br>UB1 (I will only use a BEV in the next 3 years.)<br>UB2 (I will only use a BEV in the next 5 years.) | [21] |

*3.4. Tools for Data Analysis*

The reliability and validity of the measurement model were tested using WarpPLS 7.0 for partial least squares structural equation modeling (PLS-SEM) [39]. PLS is found in many studies of technology using the UTAUT [25,47]. The latent variable reliability test criteria were Cronbach's alpha and Composite Reliability [49–51]. The aim of the validity test was to assess the construct validity in two aspects: convergent validity and discriminant validity. The structural model assessment of this research was based on two variables of decision coefficients: purchase intention (PI) and use behavior (UB) [51]. The evaluation of the structural model could be performed by using the goodness of fit criteria, which measured the relationships between latent variables. The goodness of Fit in the WarpPLS analysis included the model fit and quality indices: average path coefficient (APC), average R-squared (ARS), average adjusted R-squared (AARS), average block VIF (AVIF), average full collinearity VIF (AFVIF), Tenenhaus GoF (GoF), Simpson's paradox ratio (SPR), R-squared contribution ratio (RSCR), statistical suppression ratio (SSR) and non-linear bivariate causality direction ratio (NLBCDR) [39].

## 4. Analysis and Results

*4.1. Descriptive Statistics of Respondents and UTAUT Constructs*

The descriptive statistics of respondents are reported in Table 2. The majority of respondents were male (52.1%), were aged between 26 and 33 (34.2%), had completed graduate school (45.9%), were government officer/ employees (62%), had a monthly income in the range from THB 15,001 to 25,000 (26.6%) (USD 1 = THB 33.38 as of 30 December 2019), were living in Bangkok (70.7%), and had one car (74.2%).

**Table 2.** Descriptive statistics of survey respondents (n = 403).

| Category | Number | Percentage |
|---|---|---|
| Gender | | |
| Male | 210 | 52.1 |
| Female | 193 | 47.9 |
| Age | | |
| 18–25 | 13 | 3.2 |
| 26–33 | 84 | 20.8 |
| 34–41 | 138 | 34.2 |
| 42–49 | 91 | 22.6 |
| 50 and over | 77 | 19.1 |
| Education | | |
| Under Bachelor's degree | 33 | 8.2 |
| Bachelor's degree | 174 | 43.2 |
| Master's degree | 185 | 45.9 |
| Doctor's degree | 11 | 2.7 |
| Occupation | | |
| government officer/employees | 250 | 62.0 |
| state enterprise employees | 18 | 4.5 |
| private company employees | 80 | 19.9 |
| business owners | 31 | 7.7 |
| others | 24 | 6.0 |
| Income (THB) | | |
| less and 15,000 | 27 | 6.7 |
| 15,001–25,000 | 107 | 26.6 |
| 25,001–35,000 | 94 | 23.3 |
| 35,001–45,000 | 58 | 14.4 |
| 45,001–55,000 | 36 | 8.9 |
| 55,000–65,000 | 25 | 6.2 |
| 65,001 and over | 56 | 13.9 |
| Accommodation province | | |
| Bangkok | 285 | 70.7 |
| Nonthaburi | 62 | 15.4 |
| Samutprakan | 13 | 3.2 |
| Nakhonpathom | 12 | 3.0 |
| Pathumthani | 22 | 5.5 |
| Samutsakhon | 9 | 2.2 |
| Number of owned cars | | |
| 1 | 299 | 74.2 |
| 2 | 75 | 18.6 |
| More than 2 | 29 | 7.2 |

The descriptive statistics of all variables in the hypothesized model were computed. It was found that there was no missing information. The mean scores indicated positive results for all constructs in Table 3.

**Table 3.** Descriptive statistics of the UTAUT constructs (n = 403).

| Constructs | Minimum | Maximum | Mean | S.D. |
|---|---|---|---|---|
| Performance Expectancy | 2.00 | 5.00 | 3.86 | 0.55 |
| Effort Expectancy | 2.00 | 5.00 | 3.89 | 0.65 |
| Social Influence | 1.00 | 5.00 | 3.42 | 0.83 |
| Facilitating Conditions | 1.86 | 5.00 | 3.86 | 0.60 |
| Hedonic Motivation | 1.50 | 5.00 | 3.34 | 0.77 |
| Price Value | 1.00 | 5.00 | 3.44 | 0.71 |
| Environmental Concern | 1.00 | 5.00 | 4.20 | 0.71 |
| Policy Measures | 1.00 | 5.00 | 4.23 | 0.74 |
| Purchase Intention | 1.17 | 5.00 | 3.75 | 0.78 |
| Use Behavior | 1.33 | 5.00 | 3.68 | 0.82 |

*4.2. Measurement Model and Structural Model*

As shown in Table 4, Cronbach's alpha and Composite Reliability were used to test the quality and reliability, the values of which for all scales exceeded the minimum threshold level of 0.70 [50]. All the scales had a score above 0.7. Regarding convergent validity, the square root of the average variation extract (AVE) of all the values exceeded the minimum threshold level of 0.50 [51]. Eight of the constructs had a score above 0.5, but two of the constructs (performance expectancy, facilitating conditions) had scores of less than 0.5. A Composite Reliability value of the two constructs have above 0.6 is at an acceptable level [50]. For discriminant validity, ten constructs revealed relatively high variances extracted for each factor compared to the interscale correlations [50].

**Table 4.** Reliability, and convergent, and discriminant validity of measurement model.

| Constructs | Composite Reliability | Cronbach's Alpha | AVE | Discriminant Validity |
|---|---|---|---|---|
| PE | 0.866 | 0.826 | 0.397 | 0.630 |
| EE | 0.917 | 0.879 | 0.736 | 0.858 |
| SI | 0.911 | 0.883 | 0.632 | 0.795 |
| FC | 0.848 | 0.790 | 0.454 | 0.674 |
| HM | 0.891 | 0.837 | 0.672 | 0.820 |
| PV | 0.848 | 0.742 | 0.624 | 0.790 |
| EC | 0.930 | 0.899 | 0.768 | 0.876 |
| PM | 0.937 | 0.921 | 0.679 | 0.824 |
| PI | 0.932 | 0.911 | 0.696 | 0.834 |
| UB | 0.883 | 0.799 | 0.716 | 0.846 |

PE, Performance Expectancy; EE, Effort Expectancy; SI, Social Influence; FC, Facilitating Conditions; HM, Hedonic Motivation; PV, Price Value; EC, Environmental Concern; PM, Policy Measures; PI, Purchase Intention; UB, Use Behavior; AVE, Average Variance Extract.

The model fit and quality indices were used to test the overall results, as shown in Table 5. The *P* values for the average path coefficient (APC), average R-squared (ARS), and average adjusted R-squared (AARS) were equal to or lower than 0.05 [39]. Both the average block VIF (AVIF) and average full collinearity VIF (AFVIF) are equal to or lower than 5 [52]. The value of Tenenhaus GoF (GoF) is small if equal to or above 0.1, medium if equal to or above 0.25 and large if equal to or above 0.36 [53]. In the meantime, other indicators, the Simpson's paradox ratio (SPR), R-squared contribution ratio (RSCR), statistical suppression ratio (SSR) and non-linear bivariate causality direction ratio (NLBCDR) were equal to or above 0.7, 0.9, 0.7 and 0.7, respectively [39]. All the results met the suggested value, indicating that the model has good structural fitting. Moreover, the model had an explanatory power of 55.7% in use behavior and 58.5% in purchase intention (Table 6).

**Table 5.** Model fit and quality indices for the final model.

| Measure | Value | *P*-Values |
|---|---|---|
| Average path coefficient (APC) | 0.179 | *P* < 0.001 |
| Average R-squared (ARS) | 0.576 | *P* < 0.001 |
| Average adjusted R-squared (AARS) | 0.571 | *P* < 0.001 |
| Average block VIF (AVIF) | 1.931 | acceptable if ≤5, ideally ≤3.3 |
| Average full collinearity VIF (AFVIF) | 2.159 | acceptable if ≤5, ideally ≤3.3 |
| Tenenhaus GoF (GoF) | 0.606 | small ≥0.1, medium ≥0.25, large ≥0.36 |
| Simpson's paradox ratio (SPR) | 0.900 | acceptable if ≥0.7, ideally = 1 |
| R-squared contribution ratio (RSCR) | 0.996 | acceptable if ≥0.9, ideally = 1 |
| Statistical suppression ratio (SSR) | 1.000 | acceptable if ≥0.7 |
| Non-linear bivariate causality direction ratio (NLBCDR) | 1.000 | acceptable if ≥0.7 |

**Table 6.** Coefficient of Determinant R-Squared.

| | $R^2$ | $R^2$ **Adjusted** |
|---|---|---|
| PI | 0.593 | 0.585 |
| UB | 0.559 | 0.557 |

PI, Purchase Intention; UB, Use Behavior.

### 4.3. Hypothesis Testing

Figure 2 and Table 7 indicates the results of the structural model and their path coefficients, which showed positive direct effects on the UTAUT constructs. Six hypotheses were supported, namely, H1, H2, H3, H5, H7 and H9, which were all significant at the 1% level. Performance expectancy (PE), effort expectancy (EE), social influence (SI), hedonic motivation (HM), and environmental concern (EC) had a significant positive effect on BEV purchase intention (PI), with a standardized path coefficient (0.19, 0.14, 0.12, 0.22 and 0.22, respectively). Additionally, BEV purchase intention (PI) had a prominent effect on use behavior (UB), with a standardized path coefficient (0.75). The results indicate that there is a strong positive relationship between purchase intention and use behavior towards BEVs. However, three hypotheses were not supported. H4a and H4b: facilitating conditions (FC) did not have a significant positive effect on BEV purchase intention (PI) and user behavior, with a standardized path coefficient (0.03 and 0.08, respectively). Finally, H6 and H8: price value (PV) and policy measures (PM) were not valid, with a standardized path coefficient (0.01 and 0.08, respectively).

**Table 7.** Path results of research model.

| Hypotheses | Path Coefficient | *P*-Value | Results |
|---|---|---|---|
| H1. → PE PI | 0.19 | *** | Supported |
| H2. → EE PI | 0.14 | ** | Supported |
| H3. → SI PI | 0.12 | ** | Supported |
| H4a. → FC PI | 0.03 | 0.25 | Not Supported |
| H4b. → FC UB | 0.08 | 0.07 | Not Supported |
| H5. → HM PI | 0.22 | *** | Supported |
| H6. → PV PI | 0.01 | 0.42 | Not Supported |
| H7. → EC PI | 0.22 | *** | Supported |
| H8. → PM PI | 0.08 | 0.06 | Not Supported |
| H9. → PI UB | 0.75 | ** | Supported |

*** *P* < 0.001, ** *P* < 0.01.

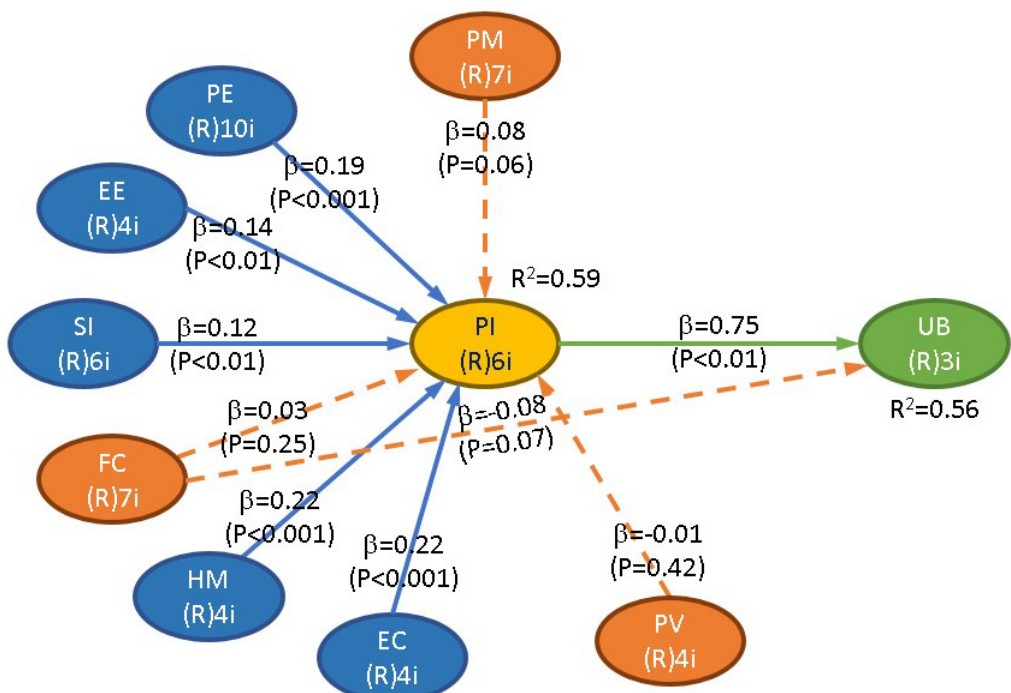

**Figure 2.** The results of the research model. PE, Performance Expectancy; EE, Effort Expectancy; SI, Social Influence; FC, Facilitating Conditions; HM, Hedonic Motivation; PV, Price Value; EC, Environmental Concern; PM, Policy Measures; PI, Purchase Intention; UB, Use Behavior.

Additionally, the effects of socio-demographic variables on the purchase intention and use behavior for BEVs were determined in multi-group analyses. However, only the age factor was found to have significant differences. Respondents were categorized into five groups for age: Group 1: 18–25 years old; Group 2: 26–33 years old; Group 3: 34–41 years old; Group 4: 42–49 years old; and Group 5: 50 years old and over. We found a significant affect on the relationship between this factor and purchase behavior as follows:

Groups 2 (26–33 years old) and 3 (34–41 years old) policy measures and environmental concerns, with a standardized path coefficient (0.27and 0.26, respectively).

Groups 3 (34–41 years old) and 4 (42–49 years old) environmental concern and social influence, with a standardized path coefficient (0.32 and 0.24, respectively).

Groups 3 (34–41 years old) and 5 (50 years old and over) performance expectancy, with a standardized path coefficient (0.22).

Groups 4 (42–49 years old) and 5 (50 years old and over) performance expectancy, with a standardized path coefficient (0.27).

### 4.4. Policy Measures to Support BEV Adoption

In the final part of the questionnaire, participants were asked to give suggestions about policy measures that they thought Thailand should implement for BEV adoption, as shown in Figure 3. The majority of respondents (359; 88.6%) suggested the exemption of car tax. The second most frequent suggestion (281 respondents; 69.7%) was free charging. The third most frequent suggestion (241 respondents; 59.8%) was subsidies or discounted car prices. Furthermore, 166 respondents (41.2%) suggested increased car tax emissions; 158 respondents (39.2%) suggested reduced or no toll fees; 149 respondents (37%) suggested an environmentally friendly city area; 145 respondents (36%) suggested eliminating the use of combustion cars in the future; and 142 respondents (35.2%) suggested subsidized or free parking fee, respectively. Moreover, 60 respondents (14.9%) suggested others such as increasing the number of charging stations, subsidies maintenance cost, supporting domestic research and development in EV technology, implementing EV law and regulation regarding infrastructure and waste, and collaboration between departments.

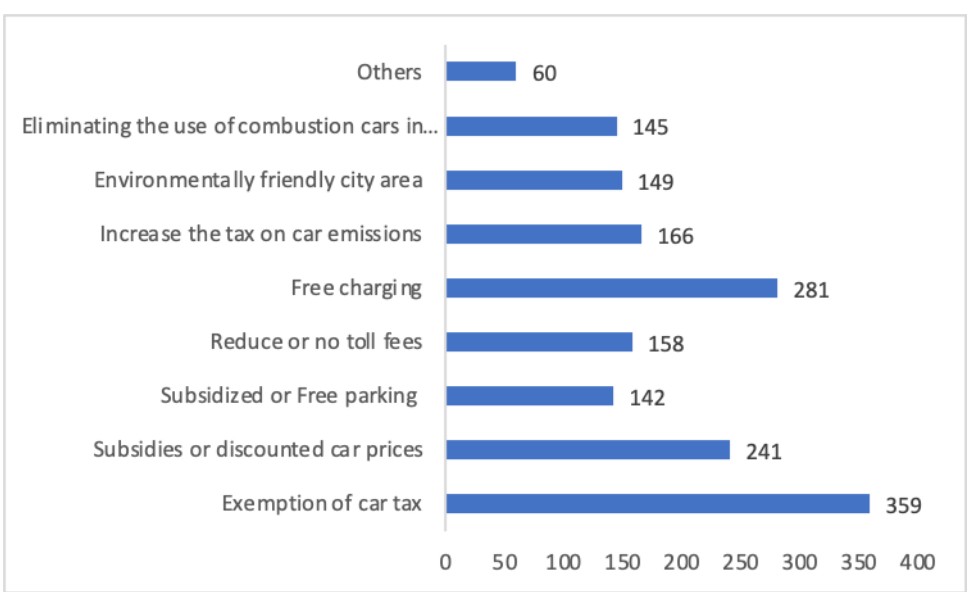

**Figure 3.** The results of policy suggestions.

## 5. Discussion of the Obtained Results

The empirical results demonstrate that the extended UTAUT2 theoretical model was suitable in the context of car drivers adopting BEVs. The adjusted $R^2$ values were 58.5% for purchase intention and 55.7% for use behavior, which both exceeded the recommended values. Among all the constructs, performance expectancy (PE), effort expectancy (EE), social influence (SI), facilitating conditions (FC), hedonic motivation (HM), price value (PV), environmental concern (EC), policy measures (PM), purchase intention (PI), and use behavior (UB), supporting H9 and H6, showed positive effects on purchase intention and use behavior. H1 suggested a positive association between performance expectancy and purchase intention. Performance expectancy has an influence on purchase intention in new technology [23,54]. Battery electric vehicles have a limited distance, which concerned the respondents the most. To support future purchases of battery electric cars, suppliers must develop the efficiency of electric cars. H2 suggested a positive association between effort expectancy and purchase intention. Effort expectancy had a major impact on purchase intention [55]. Users would prefer a battery electric car that is easy to use. H3 suggested a positive association between social influence and purchase intention. Social aspects can influence human behavior and purchase intention [23]. H5 suggested a positive association between hedonic motivation and purchase intention. This result is in accordance with the UTAUT model from previous studies [23,56]. Electric vehicle technology drives hedonic consumption among individual users of electric cars. H7 suggested a positive association between environmental concern and purchase intention. This is in accordance with studies in the same context of hybrid purchasing [39]. Nowadays, the environmental and social concerns have embraced social media in the form of campaigns and movements locally and globally. Many social media channels can be used for the use of battery electric cars instead of combustion cars. Moreover, social influencers who have numerous followers and huge credibility can build environmental awareness and support the use of battery electric cars by reviewing battery electric cars on their channels. In Thailand, initiatives have mainly concentrated on government sectors: the use of electric cars for government agencies, before focusing on residents. H9 suggested a positive association between purchase intention and use behavior. Purchase intention was confirmed to have a significant influence on use behavior in relation to battery electric cars [25].

Facilitating conditions had no significant impact on purchase intention and use behavior. H4a and H4b rejected a positive relationship between facilitating conditions and purchase intention as well as use behavior. The influence of facilitating conditions on purchase intention and use behavior became significant when users had experience with

electric vehicles [57]. Participates were concerned about the necessary resources for battery electric vehicles, such as charging stations and service centers. Electric cars are a new technology in Thailand so the facilities are limited, especially charging stations. The number of people using electric cars is relatively small. Most participants had no experience with using battery electric vehicles. The price of the cars influences whether someone buys an electric car or not [58]. H6 rejected a positive relationship between price value and purchase intention. The price of electric cars is higher than that of combustion cars at present. The government plans to implement measures to subsidies or discount car prices to increase purchases of electric cars. Finally, H8 rejected a positive relationship between policy measures and purchase intention. The incentive policies, including purchasing subsidies, convenience measures and increased charging facilities, have an important influence on the purchase intention and use behavior of using electric vehicles [30]. The Thai government has developed incentives to promote the use of electric cars, but the policy measures are unstable. People become uncertain, even if the same incentive policies will differ to their different psychological perceptions.

There were significant differences between age groups in relation to the purchase behavior of battery electric cars. The results suggest that older people might choose battery electric cars that are more efficient. Therefore, car manufacturers must improve car performance and technology. People aged 34–41 are more willing to pay for environmental protection and social influence than older generations. This implies that environmentally friendly battery electric vehicles can be promoted via social influence. Moreover, the young generations pay attention to policy measures for purchase intention.

Practically, the government, car manufacturers and the private sector should raise awareness of the importance of battery electric cars, such as environmental impact and reducing energy use. Car manufactures should speed up the development of electric cars to improve their performance, reduce costs, supply services, and advertise their cars. Furthermore, the government should measure monetary and non-monetary policies of electric cars to promote consumers' purchasing intention and actual behavior as well as make an overall plan for charging resources and stations to raise the consumption potential of battery electric cars in the future.

## 6. Conclusions

Battery electric car adoption among Thai drivers has not been thoroughly investigated in the previous literature. To bridge this gap, this study characterizes factors using the expanded UTAUT model by adding policy measures, hedonic motivation, price value, and environmental concerns. The results suggest that the research model has good explanatory power. Performance expectancy, effort expectancy, social influence, hedonic motivation, and environmental concern were significant factors of purchase intention. On the contrary, facilitating conditions, price value, and policy measures were not found to have a significant effect on purchase intention. Thai drivers expect high performance, the availability of charging facilities, ease and convenience of use, safety, cost savings, environmental protection, and enjoyment when using battery electric cars.

The limited distance and charging stations are significant disadvantages of using battery electric cars. In addition, the prices of battery electric vehicles and their parts are still much higher than those of vehicles with engines. This is a solid barrier to motivating consumers to use battery electric vehicles. Purchase intention was reflected by respondents' actual behavior towards battery electric cars. Moreover, facilitating conditions is not a significant factor in user behavior. The impact of performance expectancy, social influence, environmental concern, and policy measures on purchase intention is moderated by age.

Policy measures are necessary to attract more users, including the exemption of car tax, subsidy policies, and free charging and tolls. The government must develop necessary laws and regulations for electric cars and facilities. Moreover, relevant organizations in the public and private sectors should invest more in electric vehicle studies and research and development. Examples include how the efficiency of battery electric vehicles can

be improved, how to use battery electric vehicles within organizations, and how to make preparations to manage charging stations and prepare facilities to support EVs. In fact, the perception of this technology is constantly changing over time. These changes can possibly lead to different conclusions.

## 7. Limitations and Future Directions

This study has some limitations that should be explored in further research. Firstly, the sample included owners of combustion cars because a small number of people use battery electric cars. It was difficult to gather real users in this study. For this reason, future research should focus on existing battery electric car users to study the influencing factors of the actual purchasing behavior of battery electric car users. Moreover, other parts of Thailand should be examined to expand and potentially reaffirm our findings. Second, purchase intention and use behavior could be affected by various factors other than those mentioned in this study. Future research should include some new factors, such as use experience, fuel efficiency, and brand loyalty.

**Author Contributions:** Conceptualization, P.M. and K.C.; methodology, P.M. and K.C.; formal analysis, P.M.; investigation, P.M.; resources, P.M.; data curation, P.M.; writing—original draft preparation, P.M. and K.C.; writing—review and editing, P.M. and K.C.; visualization, P.M.; project administration, P.M.; funding acquisition, P.M. All authors have read and agreed to the published version of the manuscript.

**Funding:** This research was funded by the Energy Policy and Planning Office (EPPO), Ministry of energy, Thailand.

**Institutional Review Board Statement:** The study was conducted in accordance with the Declaration of Helsinki, and approved by the Ethics Committee of Research Involving Human Subjects, the second Allied Academic Group in Social Sciences, Humanities and Fine and Applied Arts at Chulalongkorn University. (COA No.250/2564; date of approval: 7 October 2021).

**Informed Consent Statement:** Informed consent was obtained from all subjects involved in the study.

**Acknowledgments:** Authors are grateful for the financial support from the Energy Policy and Planning Office (EPPO), Ministry of energy, Thailand.

**Conflicts of Interest:** The authors declare no conflict of interest.

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
