# Peer review of "Factors Influencing Battery Electric Vehicle Adoption in Thailand—Expanding the Unified Theory of Acceptance and Use of Technology’s Variables"

_sustainability, doi:10.3390/su14148482_

Round 1
Reviewer 1 Report
The paper describes a hypothesis testing about factors influencing EV adoption in Thailand, and in particular in citizens of Bangok. While results presented have an obvious limitation, due to the territorial context which they apply to, I think that the paper provides good insights and add on present literature on the thematic concerned. Moreover, the statistical approach is appropriate.
My major concern is about English language and syntax, which is currently very poor. Moreover, I think that data could be explained: organize information in bullet points when this is needed; avoid not useful repetition; try to present statistical indicators in an intuitive and proper way.
Author Response
We would like to thank the reviewer for the review and suggestion.
Please see the attachment.

Reviewer 2 Report
The work is very good and relevant, just need to update the references for the years 2020 and 2021.
Author Response
We would like to thank the reviewer for the review and suggestion. Please see the details below.
Point 1: update the references for the years 2020 and 2021
Response 1: We have already updated the references for the years 2020 and 2021 as follows:
- Macioszek, E. (2020). Electric Vehicles - Problems and Issues: 169-183.
- Ling, Z., Cherry, C. R., Wen, Y. (2021). "Determining the Factors That Influence Electric Vehicle Adoption: A Stated Preference Survey Study in Beijing, China." Sustainability 13(21): 11719.
- Yang, C., Tu, J.- C., Jiang, Q. (2020). "The Influential Factors of Consumers’ Sustainable Consumption: A Case on Electric Vehicles in China." Sustainability 12(8): 3496.
From the references, we add more information that relates to and benefit our research, such as an overview of factors influencing battery electric vehicle adoption, problems, and issues.
Reviewer 3 Report
The purpose of the reviewed paper is study in order to examine factors that influence Thailand car owners’ adoption of battery electric vehicle. The proposed model has been developed factors from expanding the Unified Theory of Acceptance and Use of Technology (UTAUT) as the theoretical framework. Data were collected by an online questionnaire survey completed by participants in Bangkok and the vicinity and analyzed using partial least squares structural equation modeling (PLS-SEM). The result showed that purchase intention is significantly and positively influenced by performance expectancy, effort expectancy, social influence, hedonic motivation, and environmental concern. Contrarily, purchase intention is not significantly influenced by price value and policy measures. In my opinion, the paper can be published, after taking into account the following remarks:
- the paper title should be improved because the present form is not understandable for a wide audience because the Authors used the acronym "UTAUT’s variables". It should be improved (used the full name of this acronym, or just to formulate anew the paper title),
- the paper text should be formatted according to the Sustainability journal paper template requirements. As far now, many places in the paper do not fulfill these requirements. It should be improved,
- at the end of the Introduction section, the Authors wrote what was the main aim of the paper. It is very good, but the Authors should also shortly write what was contained in each paper section,
- before paper publishing, the extensive English checking is desirable (be a professional Native Speaker),
- The subject of this paper is the problem of factors influencing battery electric vehicle adoption in Thailand. In the Introduction section, among others, the Authors presented the basic information and characteristics regarding the problems connected with the introduction of electric vehicles on the market, types of electric vehicles, and some information regarding the policies to support battery electric vehicle adoption in order to achieve the sustainable development goals, etc. It is very good, but the Authors should also add some characteristics of electric vehicles because this paper is dedicated to them. Thus, the Authors should refer to the latest research papers and research work in this area, e.g., like: "Electric vehicles - problems and issues", doi 10.1007 / 978-3-030-35543-2_14; "Determining the Factors That Influence Electric Vehicle Adoption: A Stated Preference Survey Study in Beijing, China", doi.org / 10.3390 / su132111719; "E-mobility infrastructure in the Górnoslasko - Zaglebiowska Metropolis, Poland, and potential for development", doi 10.11159 / icert19.108. One short paragraph in the Introduction section will be enough,
- it would be good to add (as section 2) a literature review on the subject of factors influencing battery electric vehicle adoption on the example of various countries around the world. This article is missing such an overview,.
- the structure of the article should be refined. It cannot be that a subsection consists only of 2-3 sentences. It is unacceptable in serious scientific papers. This is the case, a short subsection exists for example 2.2., 2.3., 2.4, 2.5, 2.6, 2.7, 2.8, 2.9, 2.10., And others. Should be corrected,
- some literature reviews can be found in the section called "2. Research Model and Research Hypotheses",
- the content of the article should be organized according to the following scheme: Introduction, Literature review, Materials and Methods (together with research hypotheses), Analysis and Results, Discussion of the obtained results and Conclusions,
- line 206: the Author wrote as follows: ..."The number of required samples is 400 respondents"... Do the Authors check if this sample number is enough from the statistical point of view and fulfill the requirement of minimum sample data set for further conducting analysis?
- is "6. Conclusion" should be "6. Conclusions",
- the Conclusion section is written in a very general way and should be extended with the presentation of detailed conclusions from the research described in this paper.
Author Response

(The authors gave the same response as above.)
